# Psychometric evaluation of the Chinese version of the Person-Centred Care Assessment Tool

Cai Le [1], Ke Ma,[2] Pingfen Tang,[3] David Edvardsson,[4] Lina Behm,[5] Jie Zhang,[2] Jiqun Yang,[6] Haiyan Fu,[6] Gerd Ahlström [7]

## ABSTRACT

**Objective** This study aims to test a Chinese cross-cultural adaptation of the English version of the Person-Centred Care Assessment Tool (P-CAT) and evaluate its psychometric properties.

**Design** P-CAT was translated/back-translated using established procedures before the psychometric evaluation of the Chinese version was made.

**Setting** Two hospitals covering urban and suburban areas of Kunming in the Yunnan province of China.

**Participants** 152 female hospital staff completed the survey.

**Main outcome measure(s)** Construct validity and reliability, including internal consistency and test–retest reliability, were assessed among a sample of hospital staff.

**Results** The factor analysis resulted in a two-component solution that consisted of two subscales. The corrected item-total correlations for all of the items ranged from 0.14 to 0.44, with six items not meeting the cut-off level for item-total correlation (>0.3). The Chinese P-CAT demonstrated strong reliability, with a Cronbach's alpha of 0.91–0.94 for the scales and a test–retest reliability coefficient of 0.88 for the overall scale scores. The intraclass correlation was 0.92 (95% CI 0.90 to 0.95).

**Conclusion** P-CAT appears to be a promising measure for evaluating staff perceptions of person-centredness in Chinese hospital environments. The results show that P-CAT can be a useful tool for improving the quality of healthcare in terms of person-centred care in the Chinese context.

### Strengths and limitations of this study

► A previous study found the need for revision of the original Person-Centred Care Assessment Tool (P-CAT).
► This study validates a translated and culturally adapted but otherwise unmodified English version of P-CAT in a Chinese context.
► The study has high response rate (92.1%).
► Convenience sampling method may limit the ability to generalise the results.
► The Chinese P-CAT has been tested only in this geriatric hospital setting.

For numbered affiliations see end of article.

**Correspondence to**
Professor Gerd Ahlström;
gerd.ahlstrom@med.lu.se

## INTRODUCTION

Population ageing is occurring globally, and multimorbidity is highly prevalent, particularly in older people, who are generally frequent healthcare users.[1] The Chinese population is one of the fastest ageing populations in the world, and the number of people aged 60 years and older is expected to rise from 193 million (13.9%) in 2013 to 454 million (32.8%) in 2050.[2] This rapid increase in the number of older people in China is reflected in the increasing number of older people admitted to hospitals. The National Health Services Survey of China shows that the hospitalisation rate of people aged 65 years and older increased significantly from 6.1% to 19.9% during the period 2003–2013,[3] and chronic diseases occur more frequently in older people and are more difficult to cure when multiple diseases coexist.[4] Thus, the healthcare system in China is facing great challenges due to the ageing population, who requires high-quality care from a wide range of multiple care professionals.

Person-centred care (PCC) is now regarded as a valuable approach to use when it comes to improving long-term care for older people,[5] and the quality of care is often described in terms of it being person-centred.[6–8] The growing interest in applying PCC is accompanied by an interest in developing instruments, commonly questionnaires answered by staff as a proxy for the ill person.[7 9] The core value of PCC is the social relationship between the staff and the ill person and the close family, characterised by respect for the person's will, mutuality, openness, honesty and sympathetic presence.[8] Even though there is a range of terms in the literature describing PCC, all imply a similar philosophy of looking at the person's needs instead of a disease.[7] A review shows that person-centred interventions are multifactorial and that the complexity of the interventions can explain the range of existing outcome variables.[10]

To date, a number of instruments have been developed to assess PCC in residential care facilities for older people, and these instruments are mainly based on the opinions of staff working in such facilities. These instruments are the English Person-Centred Climate Questionnaire-Staff version,[11] the Person-Centred Care Assessment Tool (P-CAT),[12] the Staff Assessment Person-Directed Care tool,[13] the Individualized Care tool,[14] and the Staff Person-Centred Practices in Assisted Living tool.[15] Of these, the tool most commonly used internationally, with established credibility, is the P-CAT.[7 9] The P-CAT questionnaire was designed by Edvardsson and colleagues[12] and involves a caregiver-based, self-report assessment scale used to measure the extent to which the staff agree that a number of predefined characteristics/ processes of PCC are in place within their team, unit or nursing home. P-CAT is easy to use, with only 13 items. It has a process focus, as evidenced in items connected with work routines, residents' life stories, assessment of personal needs and discussion concerning best care for the residents. It has been validated and used in English,[12] Swedish,[16 17] Korean,[18] Spanish,[19] Norwegian[20] and Chinese.[21]

A recently published literature review[22] indicates that studies of PCC were predominantly on populations in Western Europe and the USA. However, the increasing proportion of older people implies an urgent need for the implementation of PCC for the purpose of improving the quality of care. The study of Zhong and Lou[21] found that P-CAT was a good tool to adapt for use in the Chinese context. A 24-item Chinese version of the original English 13-item P-CAT was developed and later reduced to 15 items as a result of psychometric validation on staff. This modified version of P-CAT was changed to such an extent as to make it impossible to compare the results with results obtained by use of the original English version. Two items were removed and four items were added, generating three scales.[21] This Chinese version was validated on staff working in residential care for patients with dementia, since P-CAT was originally intended for use as a research instrument to measure the presence of PCC in this context.[12 23]

Palliative care is provided in a person-centred environment, where the older person and the family feel welcome, feel that they are seen and feel that they are involved by the staff. This type of supportive care has been developed in hospice settings by specialists in the care of patients dying from incurable cancer, and older persons dying from other diseases do not have the same access to palliative care.[24 25] The ageing people with multiple morbidities have complex care needs and multiple symptoms in their last period of life, which requires both geriatric and palliative expertise. However, staff need education about palliative care to be able to provide PCC integrated with a focus on symptom management and quality of life for the older people and their family.[24 26]

To date, palliative care is widely provided in Western countries, but access to such service is extremely limited in mainland China,[27] especially outside oncology departments. Since palliative care is still in its initial stage and not enough in respect of the ageing population and the number of patients with life-threatening diseases that are increasing in China,[27 28] the intervention project entitled 'Palliative care for elderly people with chronic illnesses: a comparative study between Sweden and China' was initiated. The purpose of the project is to improve palliative care through workplace education in geriatric hospital care.[29] In order to promote the improvement of quality of care through implementation of person-centred palliative care in a Chinese healthcare context, further testing and cross-validation of the original P-CAT were performed on staff in Chinese geriatric palliative hospital settings. A short instrument as P-CAT was preferable in this study due to the need for combinations with additional measurements. It is necessary to ensure that this outcome measure is applicable in the evaluation of educational intervention for staff working in a geriatric hospital. Also, a psychometric evaluation of the original 13 items of the English version means that it should be possible to compare results from Sweden in forthcoming studies within the current intervention project involving Sweden[30] and China.

In China, PCC remains a poorly understood approach that is not generally applied in geriatric services, and one of the possible barriers is the absence of a dependable and valid instrument to measure PCC. Therefore, the aim of this study was to accomplish a cultural adaptation of the original English version of P-CAT for Chinese nurses, as well as evaluate the psychometric properties of the newly translated Chinese version applied in geriatric palliative care.

## METHODS

This study tested and evaluated the Chinese version of P-CAT in Mandarin, the most common language in China, on nurses in a hospital-based palliative care setting. This psychometric study is part of the intervention project 'Palliative care for elderly people with chronic illnesses: a comparative study between Sweden and China'.

### The original English instrument

P-CAT measures the extent to which staff engaged in long-term care of the elderly consider their settings to be person-centred. The English P-CAT questionnaire has been validated in an Australian sample of nursing home staff and was shown to have satisfactory psychometric properties.[12] The self-reported questionnaire exists in both a staff version and a version for older persons. In this study, the version for staff was used. It comprises 13 items covering three subscales (the extent of personalising care, the amount of organisational support and the degree of environmental accessibility). The extent of personalising care was measured through items 1–7, amount of organisational support was measured through items 8–11, and degree of environmental accessibility was measured

through items 12–13. A 5-point Likert-type scale ranging from 1 (strongly disagree) to 5 (strongly agree) was used for scoring. Items 8–12 were negatively worded and the responses were reverse-scored before analysing the data and calculating a total score. Aggregated scores were calculated by means of simple sum scores at the subscale and total scale levels. The total score ranged from 13 to 65 and a higher score indicated a higher degree of person-centredness.

### Translation and cross-cultural adaptation of P-CAT

The process of translation and cross-cultural adaptation followed the International Test Commission guidelines.[31 32] Three native Chinese speakers produced independent forward translations from English to Chinese; two had a public health degree and the third was a senior nurse who had worked in a department of palliative care for over 10 years. First, the translators agreed on a consensus version after discussing the different versions. Then this consensus version was back-translated into English by two new translators who were bilingual and had no knowledge of the procedures of the forward translation. After that, any discrepancies were resolved by an expert committee composed of all the translators, two geriatric physicians, one senior nurse and two university professors. The back-translated version was modified by comparing it with the original version. The expert committee culturally adapted a few phrases to fit the Chinese setting, resulting in a prefinal version. The final version of the Chinese P-CAT was generated after pretesting for face validity, involving 10 female nurses from a municipal hospital in Kunming, with a mean age of 31.2 years (SD ±8.3). The pretesting resulted in no changes. The final Chinese version of P-CAT is shown in the online supplementary appendix. The 10 staff members who participated in the face validity testing of the prefinal version did not participate in the later study.

### Sample, participants and data collection

The sample was selected to cover hospital nurses working with palliative care both in urban and suburban areas of Kunming, the capital of the Yunnan province in southwest China. Two municipal hospitals in Kunming were chosen through a convenience sampling method: one hospital from the Wu Hua district (an urban area of Kunming) and one from the Guan Du district (a suburban area of Kunming).

Patients who were admitted to the sampled hospitals had multiple, complex and overlapping chronic diseases, and most patients were aged 60 years or older. The following criterion for inclusion was used: municipal-level hospital with a department of palliative care.

All nurses (n=165) on duty (all shifts on the day of the data collection) at the palliative care departments of the two hospitals were deemed to be eligible for participation and were asked whether they would be willing to participate in the study. The sample size was determined by the number of available nurses working at the palliative care departments.

Finally, 152 nurses gave their informed consent and completed the Chinese version of the P-CAT questionnaire. The participants were asked to take part in both the test and retest assessments, and all 152 participants completed the first and the follow-up test. This gives an overall response rate of 92.1%. The retest was conducted between 1 and 2 weeks after the initial test.

The questionnaires were distributed to all participants by two graduate students, and the completed questionnaires were then anonymously collected on site. In addition to the P-CAT scale, demographic characteristics of the participants were also collected by means of a questionnaire that assessed age, gender, level of education, ethnicity and position.

### Psychometric evaluation

Only one item in one questionnaire had a missing value and it was replaced with the mean value of the item for the entire group.[33] An exploratory factor analysis (EFA) with extraction of principal components by both varimax and oblique rotation was used to evaluate construct validity. After either direct varimax rotating or oblique rotating, components in the structural matrix comprised the same numbers of items and loadings, indicating that no difference was found between these two methods in the analysis, and thus only the results of the analysis with varimax rotation are presented. Bartlett's test of sphericity and the Keiser-Meyer-Olkin (KMO) measure were applied in order to check the appropriateness of the factor analysis. In order to assess whether the correlation between items was adequate based on a criterion of p<0.0001, Bartlett's test of sphericity and a criterion of KMO ≥0.70 were used to indicate sample adequacy. When the Kaiser criterion of eigenvalues was ≥1, factors were extracted. A cut-off of 0.50 was applied to determine item loadings on each individual component.[34]

The internal consistency and test–retest reliability were evaluated. The internal consistency for the total and subscale scores was estimated using Cronbach's alpha coefficient, and Cronbach's alpha ≥0.50 was used as acceptable internal consistency reliability.[35] The item-total correlations were also computed, and the cut-off score for acceptable item-total correlations was set to between 0.3 and 0.8 to ensure moderate correlation and avoid item redundancy.[33] Test–retest reliability was examined with the intraclass correlation (ICC) and Pearson's correlation coefficient (r), where an ICC >0.80 was taken to indicate satisfactory reliability.[36] The paired t-test (two-tailed) was used to verify whether the mean scores from the test and retest differed.

The results of the principal component analysis (PCA) indicate that three factors with eigenvalues above 1 produced a three-component rotated solution that explained 60.2% of the total variance in the data. After direct varimax orthogonal rotating, the structure matrix showed that the first component comprised seven items,

the second comprised four items, and the third comprised two items. With regard to the internal consistency reliability, the Cronbach's alpha coefficient of the 13-item Chinese P-CAT was 0.65 for the total scale, 0.67 for the first subscale, 0.85 for the second subscale and 0.42 for the third subscale. Since subscale 3 failed to meet the reliability cut-off of 0.5, the analysis was rerun by forcing the EFA into two factors.

The two-factor model was also evaluated by means of a confirmatory factor analysis (CFA) within the framework of structural equation modelling (SEM).[37] The goodness of fit was evaluated using indices of the normed fit index (NFI), the comparative fit index (CFI) and the root mean square error of approximation (RMSEA). The results indicated that the goodness of fit of the questionnaire was 0.91 for the NFI, 0.91 for the CFI and 0.73 for the RMSEA. Thus, the CFA supported the exploratory findings. Statistical significance was determined based on two-tailed p values of <0.05. All data analyses were performed using SPSS V.22.0 software, and the two-factor model was evaluated by means of a CFA within the framework of SEM with SPSS V.22.0 under the AMOS (Analysis of Moment Structures) package.

### Patient and public involvement

Patients, relatives and staff from palliative care were involved in the development of workplace education for staff and informed by patients' priorities, experience and preferences.[29] It is planned to disseminate the results to staff at meetings about palliative care.

## RESULTS
### Demographic characteristics of the study group

As described in table 1, the sample consisted of only female nurses, with a mean age of 30.7 years (SD ±8.7, range 19–53 years). In total, 25.7% of the nurses had a bachelor's degree or higher. More than a quarter of the participants reported a minority ethnicity.

### Construct validity

Bartlett's test of sphericity was significant (754.99, p<0.0001) and the KMO measure was satisfactory (0.78), indicating that correlations between items were sufficient to conduct a PCA. As shown in table 2, the results of the PCA indicate that two factors with eigenvalues above 1 produced a two-component rotated solution that explained 50.4% of the total variance in the data. After direct varimax orthogonal rotating, the structural matrix showed that the first component comprised nine items (loadings between 0.16 and 0.79) and the second comprised four items (loadings between 0.71 and 0.89). The two subscales were labelled as (1) extent of personalising care and (2) amount of organisational and environmental support.

### Reliability

With regard to the internal consistency reliability, the Cronbach's alpha coefficient of the 13-item Chinese

**Table 1** Demographic characteristics of the study sample (N=152)

| Characteristics | n (%) |
|---|---|
| Gender | |
| Female | 152 (100.0) |
| Male | 0 (0.0) |
| Age (years) | |
| 18–30 | 96 (63.2) |
| 31–39 | 22 (14.5) |
| ≥40 | 34 (22.4) |
| Level of education | |
| High school | 14 (9.2) |
| Secondary school | 39 (25.7) |
| Junior college | 60 (39.5) |
| Bachelor | 39 (25.7) |
| Ethnicity | |
| Han | 111 (73.0) |
| Minorities | 41 (27.0) |

P-CAT was 0.92 for the total scale, 0.94 for the extent of personalised care subscale, and 0.91 for the amount of organisational and environmental support subscale (table 2), indicating that the overall internal consistency reliability was strong. The corrected item-total correlations for all items ranged from 0.14 to 0.44 (table 3), with six items (items 6, 7, 9, 11, 12 and 13) failing to meet the commonly suggested cut-off level for item-total correlation (>0.3).

Table 4 shows the results from the test–retest reliability assessment of the Chinese P-CAT. The results of Pearson's correlation coefficient analysis indicated that the Chinese P-CAT had a high correlation between the test and retest on all scale levels: on the subscales extent of personalising care (r=0.90, p<0.01) and amount of organisational and environmental support (r=0.88, p<0.01) and on the overall scale (r=0.88, p<0.01). A paired t-test also confirmed that there was no significant difference between the mean scores of P-CAT on the test and retest (p>0.05). The ICC of the total score between the test and retest was 0.92, providing further support for stating that the scale had satisfactory test–retest reliability.

## DISCUSSION

In this study, the English P-CAT was culturally adapted into a Chinese version. During the translation of the English P-CAT into Chinese, a few minor cultural discrepancies were encountered, and two items of P-CAT have been modified accordingly. In order to use words closer to the Chinese cultural context and more applicable to a hospital environment, the term 'resident' was replaced by 'patient' and the term 'free' was replaced by 'flexible'. After cross-cultural adaptation, the pretesting of the

**Table 2** Rotated component matrix for PCA of the two-factor Chinese P-CAT

| Item number | Item content | Factor loadings | |
|---|---|---|---|
| | | Subscale 1: extent of personalising care | Subscale 2: amount of organisational and environmental support |
| 1 | We often discuss how to provide person-centred care. | **0.73** | 0.008 |
| 2 | We have formal team meetings to discuss residents' care. | **0.79** | 0.045 |
| 3 | The life history of the residents is formally used in the care plans we use. | **0.76** | 0.028 |
| 4 | The quality of the interaction between staff and residents is more important than getting the tasks done. | **0.72** | −0.087 |
| 5 | We are free to alter work routines based on residents' preferences. | **0.67** | 0.117 |
| 6 | Residents are offered the opportunity to be involved in individualised everyday activities. | **0.54** | −0.109 |
| 7 | Assessment of residents' needs is undertaken on a daily basis. | **0.17** | 0.26 |
| 8 | I simply do not have the time to provide person-centred care. | −0.208 | **0.78** |
| 9 | The environment feels chaotic. | −0.096 | **0.89** |
| 10 | We have to get the work done before we can worry about a homelike environment. | **0.64** | −0.364 |
| 11 | The organisation prevents me from providing person-centred care. | −0.094 | **0.89** |
| 12 | It is hard for residents in the facility to find their way around. | −0.036 | **0.71** |
| 13 | Residents are able to access outside space as they wish. | **0.35** | −0.087 |
| Total variance explained (%) | 50.4 (total 2 subscales) | 30.18 | 20.24 |
| Cronbach's alpha | 0.92 (total 13 items) | 0.94 | 0.91 |

Bold number means included in the scale.
PCA, principal component analysis; P-CAT, Person-Centred Care Assessment Tool.

Chinese version of P-CAT revealed that the items were clearly understood.

This Chinese version of P-CAT showed a different subscale structure as compared with the original English scale, that is, a structure with three subscales (extent of personalising care, amount of organisational support and degree of environmental accessibility) consisting of 13 items. On the other hand it showed the same subscale structure as the Norwegian scale, that is, a structure which had two subscales (extent of personalising care and amount of organisational and environmental support). The 13 items were mostly in concordance with the Norwegian version, whereas only item 10 was connected to the first subscale.[20] The two-factor structure has also been confirmed in the Swedish version of P-CAT based on a large cross-sectional sample of 1465 staff members from 195 residential elderly care units[17] and in a more recent study based on 142 staff

working in residential elderly care units and 182 staff working at healthcare centres or home care centres and in social services.[16] Moreover, in this study the Chinese P-CAT showed strong internal consistency reliability with a Cronbach's alpha of 0.92 for the total scale, indicating that the Chinese P-CAT matched well with the characteristics and structure of the Norwegian version. However, the different subscale structures of this Chinese version of P-CAT as compared with the original English scale may reflect a difference in sample characteristics of staff and the cultural context. The test–retest reliability assessed by ICC was excellent for overall Chinese P-CAT score (0.92) and both subscales (0.94 and 0.91), and the excellent test–retest reliability of the Chinese P-CAT was further evidenced by a Pearson's correlation coefficient (r) of 0.90 for the total scale, 0.88 for the extent of personalising care subscale and 0.88 for the amount of

**Table 3** Item performance and reliability test of the Chinese P-CAT

| Item number | Item content | Mean±SD | Corrected item-total correlation | Cronbach's alpha if item deleted |
|---|---|---|---|---|
| 1 | We often discuss how to provide person-centred care. | 4.19±0.70 | 0.36 | 0.62 |
| 2 | We have formal team meetings to discuss residents' care. | 4.32±0.66 | 0.44 | 0.61 |
| 3 | The life history of the residents is formally used in the care plans we use. | 4.24±0.73 | 0.41 | 0.62 |
| 4 | The quality of the interaction between staff and residents is more important than getting the tasks done. | 4.50±0.65 | 0.32 | 0.63 |
| 5 | We are free to alter work routines based on residents' preferences. | 4.20±0.77 | 0.42 | 0.61 |
| 6 | Residents are offered the opportunity to be involved in individualised everyday activities. | 3.86±0.82 | 0.23 | 0.64 |
| 7 | Assessment of residents' needs is undertaken on a daily basis. | 2.30±1.10 | 0.19 | 0.65 |
| 8 | I simply do not have the time to provide person-centred care. | 3.89±0.82 | 0.35 | 0.62 |
| 9 | The environment feels chaotic. | 4.11±1.00 | 0.15 | 0.65 |
| 10 | We have to get the work done before we can worry about a homelike environment. | 1.74±0.62 | 0.37 | 0.62 |
| 11 | The organisation prevents me from providing person-centred care. | 4.14±0.92 | 0.17 | 0.65 |
| 12 | It is hard for residents in the facility to find their way around. | 2.22±0.88 | 0.26 | 0.64 |
| 13 | Residents are able to access outside space as they wish. | 3.89±0.82 | 0.14 | 0.65 |

P-CAT, Person-Centred Care Assessment Tool.

of organisational and environmental support subscale. The results of Pearson's correlation coefficient (r) for the total scale in the Chinese version were higher than in the English (0.66),[12] Swedish (0.75),[16 17] Norwegian (0.83)[20] and previous Chinese (0.68)[21] versions, which indicates that this Chinese P-CAT may have a stronger test–retest reliability than other-language versions of P-CAT.

In this Mandarin version of the Chinese P-CAT, six items (6, 7, 9, 11, 12 and 13) did not meet the commonly suggested cut-off level for item-total correlation (>0.3). In the previous Chinese version of P-CAT,[21] items 7 and 9 were deleted due to the low value of factor loading (<0.3), and four newly developed items were added to the scale.[21] Item 7 in this study also had a lower value of factor loading than is recommended. However, we decided to include this item in subscale 1, in agreement with the original English version, although this item had a slightly higher value of factor loading in the case of factor 2. Because these two Chinese versions of P-CAT were

evaluated in different healthcare settings (hospital environments and residential care facilities), there is no firm consensus regarding the psychometric properties of the Chinese P-CAT, and the findings suggest that the Chinese version requires further investigation and analysis with other samples and in other settings.

In contrast to the low response rate of 21% in the development study of the original English version of P-CAT,[12] our study showed a high response rate of 92.1%. Another strength is that the original instrument used was developed from both a clinical and a theoretical perspective on PCC that is seen as the model for good caring from an international perspective.[7 8 12] The model supports an application of the core principles of PCC in the specific sociocultural context of China. The literature suggests a need to see the framework based more on empirical studies from China.[22]

However, some limitations should be noted in this study. First, P-CAT is primarily intended for use as a research tool

**Table 4** Test–retest reliability of the Chinese P-CAT

| Scale dimension | First test (mean±SD) | Second test (mean±SD) | P value | Pearson's correlation coefficient (r) | 95% CI | ICC (95% CI) |
|---|---|---|---|---|---|---|
| Extent of personalising care | 33.35±3.59 | 33.13±3.33 | 0.09 | 0.90 | 0.83 to 0.94 | 0.94 (0.92 to 0.96) |
| Amount of organisational and environmental support | 14.41±1.72 | 14.34±1.74 | 0.34 | 0.88 | 0.76 to 0.89 | 0.91 (0.87 to 0.93) |
| Overall scale | 47.76±3.70 | 47.46±3.65 | 0.07 | 0.88 | 0.78 to 0.92 | 0.92 (0.90 to 0.95) |

P values are two-tailed p values of paired t-test.
ICC, intraclass correlation; P-CAT, Person-Centred Care Assessment Tool.

to measure the presence of PCC in residential aged care facilities, but the present study was conducted only in a hospital context, which may limit the ability to generalise the results to staff working in other healthcare contexts. Further psychometric evaluation of this Chinese P-CAT is needed within different settings to verify the results of this study. Second, from a psychometric evaluation the criterion-related validity was not assessed as a gold standard for the measurement of PCC in hospitals; however, no such criterion exists because of the lack of previous studies in hospital contexts. In addition, the discriminant validity of the Chinese version of P-CAT was not evaluated in our study. Future research should therefore investigate whether the questionnaire has the ability to demonstrate significant differences across hospitals and whether it is useful in discriminating between hospitals in terms of their caring culture. Third, the content validity of P-CAT was only considered along with the procedures of cross-cultural adaptation. Fourth, only female participants were included in this study, for which reason this study may be subject to gender bias. In interpreting the results, it should be taken into account that this result does not apply to the 2.1% male nurses in China (3.5 million nurses).[38 39] Fifth, this study used a cross-sectional design, so it would be useful to conduct a longitudinal study in the future to evaluate change over time. Sixth, the sample size despite a high overall response rate (92.1%) was limited to 152 nurses. We accepted this sample size for the possibility to compare results from the target group (nurses in palliative care) within the ongoing intervention project involving Sweden[30] and China. The sample size recommendations for factor analysis vary in the literature,[40 41] commonly 5:1[41 42] or 10:1[43] sample to item ratios. Others have suggested 100 as an absolute minimum sample size.[40 41]

Palliative care in mainland China has developed slowly since the late 1980s, but in recent years professionals, patients and their families, as well as the government, have become increasingly aware of its importance.[27 28 44] However, even though the provision of palliative care is becoming more and more common in China, the hospice and palliative care services are not meeting the growing demand from the ageing population, and the number of patients with cancer and other life-threatening diseases is increasing.[28] The use of a reliable and valid instrument for the assessment of PCC in research and practice can serve not only to enhance the quality of elderly care but also to strengthen the ongoing development of palliative care in China. Palliative care implies a person-centred approach characterised by a holistic view of the person and a belief that the person should be supported in living a life with dignity. Palliative care strives to make the whole person visible and prioritises the patient's spiritual, existential, social and psychological needs to the same extent as physical needs.[28 45 46] The patient version of P-CAT was used previously by the research group as a complementary assessment in the evaluation of palliative care interventions.[47] Another application is in the evaluation of workplace education about palliative care for staff in the nursing home context.[30] In this current project,[29 30] for instance, it is being used in order to investigate whether the staff's attitude towards person-centredness has changed as a result of education.

## CONCLUSION

The results indicated that the 13-item Chinese P-CAT, which is a cross-culturally adapted version of the English P-CAT, shows strong internal consistency and test–retest reliability and appears to be a promising measure for evaluating staff perceptions of person-centredness in a Chinese palliative hospital setting. The results of published psychometric evaluations indicate the use of two scales in P-CAT as was determined in this study. However, we suggest that the measurement properties of the Chinese version should be further evaluated in other samples and in different healthcare settings. An additional issue for future studies is to evaluate this instrument as a summative measurement with cut-off values for different levels of PCC. This can benefit healthcare staff in assessing their quality of care. The results show that P-CAT can be a useful tool for improving the quality of healthcare in terms of PCC in the Chinese context.

**Author affiliations**
[1]School of Public Health, Kunming Medical University in Kunming, Kunming, China
[2]Palliative Care, The Third People's Hospital of Kunming, Kunming, Yunnan, China
[3]School of Nursing, Kunming Medical University, Kunming, Yunnan, China
[4]School of Nursing & Midwifery, College of Science, Health and Engineering, La Trobe University/Austin Health Clinical School of Nursing, Heidelberg, Victoria, Australia
[5]Faculty of Health Sciences, Kristianstad University College, Kristianstad, Sweden
[6]Department of Palliative Care, The Third People's Hospital of Kunming, Kunming, China
[7]Department of Health Sciences, Faculty of Medicine, Lund University, Lund, Sweden

**Acknowledgements** We would like to thank Ms Huang Jingjing and Ms Wu Chao, two postgraduates of the School of Public Health, Kunming Medical University, who helped us with data collection, and Dr Magnus Persson, Lund University, for proof-reading the manuscript.

**Collaborators** KUPA-project in Sweden: Christina Bökberg, Birgit Rasmussen, Eva Benzein, Anna Sandgren.

**Contributors** CL was responsible for the study design, data analysis and drafting of the paper. GA, PT and KM contributed to the study design and provided comments on the paper during the writing process. DE and LB provided comments on the paper during the writing process. HF, JZ and JY were responsible for data collection. All authors have read and approved the final version of the manuscript.

**Funding** The study was supported by grants from the National Natural Science Fund of China (grant number: 81611130077) and the Swedish Research Council (grant number: 2015-06243).

**Competing interests** None declared.

**Patient consent for publication** Not required.

**Ethics approval** This study has been approved by the Ethics Committee of Kunming Medical University (ID number KMU01). Oral and written informed consent was obtained from all persons participating in the study.

**Provenance and peer review** Not commissioned; externally peer reviewed.

**Data availability statement** Data are available upon reasonable request. The data sets used and/or analysed during the current study are available from the corresponding author on reasonable request.

**ORCID iDs**
Cai Le http://orcid.org/0000-0001-7315-8077
Gerd Ahlström http://orcid.org/0000-0001-6230-7583

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
