## [Reviewer comments · BMJ Open]

ARTICLE DETAILS

TITLE (PROVISIONAL)	Psychometric evaluation of the Chinese version of the Person-centred Care Assessment Tool
AUTHORS	Ahlström, Gerd; Le, Cai; Ma, Ke; Tang, Pingfen; Edvardsson, David; Behm, Lina; Zhang, Jie; Yang, Jiqun; Fu, Haiyan

VERSION 1 – REVIEW

REVIEWER	Gwen McGhan University of Calgary, Canada
REVIEW RETURNED	24-Jul-2019

GENERAL COMMENTS	Thank you for the opportunity to review this paper, as it addresses an important issue, namely measuring PCC. The following are suggestions to help with the clarity of the paper: The research aim needs to be more clearly articulated. From the Background section, it is unclear exactly why the P-CAT is being translated into Chinese as it has already been validated in Chinese. How is this new tool significantly different? Is the tool being retranslated or using the translation from the previous tool? Also, there is discussion around the tool targeting Chinese geriatric hospital settings but it appears the study was conducted in geriatric palliative care. So is the tool for cross settings or palliative care? More in-depth discussion on what PCC is and how it has been measured and limitations to those measurements may also be helpful. On page 4, there appears to be circular logic re: PCC. Page 8 - what is the difference between registered nurses and enrolled nurses? Why was this particular sample targeted? Are there reasons why only nurses were included and not other healthcare providers? Page 9 - should write out PCA the first time it is used. Page 14 - Table 4: the results do not match from the text to the table In the discussion section, a more thorough examination of how the tool could be used would be helpful. Also, there is comparison to studies in residential elderly care units but the current study was conducted in a hospital setting, which limits the interpretation of the results. Lastly, it appears the study was conducted in geriatric palliative care. A description of the patient population in that context would be helpful.
---

REVIEWER	Roy Stewart University of Groningen, University Medical Center Groningen, Department of Health Sciences, Community & Occupational Medicine, Groningen, The Netherlands
REVIEW RETURNED	03-Oct-2019

GENERAL COMMENTS	Title: Psychometric evaluation of the Chinese version of the Person-centred Care Assessment Tool (P-CAT) The aim of this manuscript is “to make a cultural adaptation of the English version of the P-CAT for Chinese nurses as well as evaluate the psychometric properties”. In the introduction, the importance and necessity of this study is clearly indicated with arguments. In particular, there is a growing interest in person-centred care in China. The authors indicate that P-CAT has been evaluated and is used in English, Swedish, Korean, Spanish, Norwegian and Chinese. The conclusion of this manuscript is therefore as follows: ” P-CAT appears to be a promising measure for evaluating staff perceptions of the person-centredness in Chinese hospital environments. The results show that P-CAT can be a useful tool for management of improving quality of healthcare in terms of person-centred care in the Chinese health care context”. There are some revisions (major compulsory, and minor) to be made. Major compulsory revisions:  1. In the article summary (page 3), the authors claim that this study is the first to validate a translated version of the P-CAT. This is questionable, since there are publications such as Zhong, 2013(5) (reference 18 in this manuscript), but also Mao, 2016(1) and as mentioned in table 1 of the article by Wang, 2019(4). Could the authors indicate if the publication of Mao, 2016(1), which can only be read in Chinese, is a relevant publication? 2. The justification of the sample size for this study was based on criteria assumed by Reise et al.,2000(2) (page 8 of this manuscript). But Reise,2000(2), on p. 290, indicates that the minimum sample size can be made on the basis of different variables. For the calculation of sample size the authors can use one of the following suggestions:  1. Criteria formulated in the manuscript (paragraph: Psychometric evaluation, pp. 8-9), such as ICC > 0.8. 2. Data from evaluated translated P-CAT. For example, Zhong, 2013(5) presents different goodness or fit measures. The article from MacCallum, 1996(3) has some opportunities to do so. 3. In order to answer the research question, it was decided to use an exploratory factor analysis (EFA). In this case, it would be better to use a confirmatory factor analysis (CFA). Can the authors please indicate why a CFA was not chosen? This should be explained in the discussion if the authors continue to use an EFA. 4. The authors say on page 8 that "No difference was found between these two methods [varimax versus oblique] in the analysis...." . How was this difference calculated? 5. Can the authors please give a reference for the statement: 'the cut-off score for acceptable item-total correlations was set to between 0.3 and 0.8 to ensure moderate correlation and avoid item redundancy' page 9.
---

	Minor revisions: 6. In Table 4 the legend should be include the meaning of p or the authors should indicate at the bottom of the table the meaning of p and ICC. 7. Typo in Table 2 must be “2 subscales” and not “3 subscales”. 8. Typo on page 17 “P-CA” must be “P-CAT”. 9. The authors can also discuss the absence of women in this dataset within the limitations of the study (Table 1). 10. Multiple Imputation of missing values would be better than replacing it with the mean value (pag. 8). 11. Also an addition of the 95% confidence intervals (CI 95%) of the Pearson correlations in Table 4 could support the discussion at page 16 regarding differences between correlations. References  1. Mao P, Xiao D, Zhang M, Xie F, Feng H. Person-centered dementia care in eldercare institutions in hunan province. Journal of Nursing Sciences. 2016;31(21):1-4. 2. Reise SP, Waller NG, Comrey AL. Factor analysis and scale revision. Psychol Assess. 2000;12(3):287-297. doi: 10.1037//1040-3590.12.3.287. 3. MacCallum RC, Browne MW, Sugawara HM. Power analysis and determination of sample size for covariance structure modeling. Psychol Methods. 1996;1(2):130-149. doi: 10.1037/1082-989X.1.2.130. 4. Wang J, Wu B, Bowers BJ, et al. Person-centered dementia care in china: A bilingual literature review. Gerontol Geriatr Med. 2019;5:1-11. doi: 10.1177/2333721419844349. 5. Zhong XB, Lou VWQ. Person-centered care in chinese residential care facilities: A preliminary measure. Aging Ment Health. 2013;17(8):952-958. doi: 10.1080/13607863.2013.790925.
--	--

VERSION 1 – AUTHOR RESPONSE

Reviewers’ comments and the authors’ responses:

Reviewer 1: Gwen McGhan

Comment 1: Thank you for the opportunity to review this paper, as it addresses an important issue, namely measuring PCC. The following are suggestions to help with the clarity of the paper: a) The research aim needs to be more clearly articulated.

b) From the Background section, it is unclear exactly why the P-CAT is being translated into Chinese as it has already been validated in Chinese. How is this new tool significantly different? Is the tool being retranslated or using the translation from the previous tool?

Answer: Thank you so much for this comment. We have revised the Background section in order to clarify the rationale of the study and thereby the aim. We have also made a small change in the aim in order to make it evident that we used the original English version of P-CAT. Our reason for using the latter version was that we were not convinced that the modified version previously published in Chinese was more valid. The designer of the original English version is also one of the authors (David Edvardsson), and his expertise in the field was of great benefit in the discussions prior to the decision to choose this version. An additional factor is that if this original version shows acceptable

psychometric characteristics, we can compare our results in future studies with those from Sweden in the intervention project entitled “Palliative care for older people with chronic illnesses: a comparative study between Sweden and China” (Bökberg et al. 2019).

C2: Also, there is discussion around the tool targeting Chinese geriatric hospital settings but it appears the study was conducted in geriatric palliative care. So is the tool for cross settings or palliative care?

Answer: We appreciate your comment. Therefore we have now explained in the Background section that the rationale of this study was to validate P-CAT before use as an outcome measure in the aforementioned intervention project. The idea behind the project was to improve palliative care through work-place education in geriatric hospital care (Ahlström and Benzein 2019).

C3: More in-depth discussion on what PCC is and how it has been measured and limitations to those measurements may also be helpful. On page 4, there appears to be circular logic re: PCC.

Answer: Thank you for this comment. We have now expanded the text about PCC and how it has assessed and what limitations there were in the measurements. The sentence on page 4 has been revised.

C4: Page 8 - what is the difference between registered nurses and enrolled nurses? Why was this particular sample targeted? Are there reasons why only nurses were included and not other healthcare providers? Page 9 - should write out PCA the first time it is used.

Answer: Thank you for this important comment. In China, there are no enrolled nurses (or assistant nurses) employed by hospitals. Therefore we exclude this information in Table 1 to make it more clear for the reader. The relatives are in the hospital together with the patient and are responsible for the patient’s daily activities such as eating, bathing, getting dressed, toileting and transferring. If the relative cannot stay in the hospital, the relative privately hires a person who helps the patient instead. Also, there are commonly volunteers in Chinese hospitals who can help patients with specific tasks. All nurses in China have nowadays a bachelor’s degree but previously the nursing education differs in length between 3 years (not university education) and 4 years (university education). The work tasks are the same for all nurses. What distinguishes between them is mode of employment (contract or formal) and level of competence dependent on experience and training after their professional qualification certificate. P-CAT was developed as a self-reporting assessment scale for staff ratings of the person-centredness of their nursing practice. Since nurses are the professionals working most frequently in palliative care it was considered best to include this professional in this study.

Thanks for your good comments. We have written out PCA (principal component analysis) as you suggested.

C5: Page 14 - Table 4: the results do not match from the text to the table. In the discussion section, a more thorough examination of how the tool could be used would be helpful. Also, there is comparison to studies in residential elderly care units but the current study was conducted in a hospital setting, which limits the interpretation of the results.

Answer: Thanks for your good comments. We have revised these sentences to make sure the results match from the text to the table. We have added some sentences in the limitations section. We have also added text in the Discussion section (the paragraph before Conclusion) about P-CAT as an instrument for the evaluation of palliative care interventions in the case of patients as well as the evaluation of education about palliative care in the case of staff.

C6: Lastly, it appears the study was conducted in geriatric palliative care. A description of the patient population in that context would be helpful.

Answer: Now included. Please see Sample, participants and data collection section.

Reviewer 2: Roy Stewart

In the introduction, the importance and necessity of this study is clearly indicated with arguments. In particular, there is a growing interest in person-centred care in China. The authors indicate that P-CAT has been evaluated and is used in English, Swedish, Korean, Spanish, Norwegian and Chinese. The conclusion of this manuscript is therefore as follows: "P-CAT appears to be a promising measure for evaluating staff perceptions of the person-centredness in Chinese hospital environments. The results show that P-CAT can be a useful tool for management of improving quality of healthcare in terms of person-centred care in the Chinese health care context".

Answer: We are grateful for your judgment that the manuscript is of importance and that the introduction is written in a concise manner. However, the introduction has been expanded in the light of the comments above.

Major compulsory revisions:

C1: In the article summary (page 3), the authors claim that this study is the first to validate a translated version of the P-CAT. This is questionable, since there are publications such as Zhong, 2013(5) (reference 18 in this manuscript), but also Mao, 2016 (1) and as mentioned in table 1 of the article by Wang, 2019(4). Could the authors indicate if the publication of Mao, 2016(1), which can only be read in Chinese, is a relevant publication?

Answer: The publication of Mao, 2016 (1) assesses person-centred dementia care with the Chinese P-CAT version validated by Zhong 2013 (5). However, in Zhong's study, the original 13-item English P-CAT was translated and back-translated, and 11 items were added based on literature review and expert consultation, making the results non-comparable with the original English version, and this Chinese version was only validated in residential care facilities. This study is a part of an intervention project entitled "Palliative care for elderly people with chronic illnesses: a comparative study between Sweden and China". The hospitals included in this study in China are comparable with the nursing homes in Sweden which have medically educated staff such as registered nurses, occupational therapists, physiotherapists and physicians working on a consulting basis. This is not the case when it comes to the residential care facilities in China. One further reason to choose the original version of P-CAT was to have the possibility of a comparison between the results of a palliative care workplace education for staff in the two countries if P-CAT should show acceptable reliability and validity. Geriatric care for older people in Sweden is a question of acute care and the investigation of suspected illnesses common in old age.

C2: The justification of the sample size for this study was based on criteria assumed by Reise et al., 2000 (2) (page 8 of this manuscript). But Reise, 2000 (2), on p. 290, indicates that the minimum sample size can be made on the basis of different variables. For the calculation of sample size the authors can use one of the following suggestions:

- I. Criteria formulated in the manuscript (paragraph: Psychometric evaluation, pp. 8-9), such as ICC > 0.8.
- II. Data from evaluated translated P-CAT. For example, Zhong, 2013(5) presents different goodness or fit measures. The article from MacCallum, 1996(3) has some opportunities to do so.

Answer: We used criteria I, which we now have added on page 9.

C3: In order to answer the research question, it was decided to use an exploratory factor analysis (EFA). In this case, it would be better to use a confirmatory factor analysis (CFA). Can the authors please indicate why a CFA was not chosen? This should be explained in the discussion if the authors continue to use an EFA.

Answer: Thanks for your good comments. We performed CFA in our study according to your suggestion. Please see section Psychometric evaluation, last paragraph.

C4: The authors say on page 8 that "No difference was found between these two methods [varimax versus oblique] in the analysis...." . How was this difference calculated?

Answer: We have added some sentences to explain it in the first paragraph in the section Psychometric evaluation.

C5: Can the authors please give a reference for the statement: 'the cut-off score for acceptable item-total correlations was set to between 0.3 and 0.8 to ensure moderate correlation and avoid item redundancy' page 9.

Answer: We applied a rather commonly used cut-off from the literature to determine how closely each item correlates with the total score. The limit for satisfactory item correlation was set to >0.30. A value less than 0.30[[? of 0.30 or less]] indicates that the item does not closely correlate, whilst a value above 0.80 indicates that the item does closely correlate. The reference used was Streiner, D.L., G.R. Norman, and J. Cairney, *Health Measurement Scales: A Practical Guide to Their Development and Use*. Fifth ed. 2014, Oxford: Oxford University Press.

Minor revisions:

C6: In Table 4 the legend should be include the meaning of p or the authors should indicate at the bottom of the table the meaning of p and ICC.

Answer: We have done as you have suggested. Please see the bottom of Table 4.

C7: Typo in Table 2 must be "2 subscales" and not "3 subscales".

Answer: We have corrected the error in Table 2.

C8: Typo on page 17 "P-CA" must be "P-CAT".

Answer: We have corrected the error in the Conclusion section. Thank you for noticing this mistake.

C9: The authors can also discuss the absence of women in this dataset within the limitations of the study (Table 1).

Answer: Done as you have suggested.

C10: Multiple Imputation of missing values would be better than replacing it with the mean value (pag. 8).

Answer: Thank you for your very good suggestion. Since only one item in one questionnaire had a missing value, it was replaced with the mean value of the item for the entire group. We will use the method you suggested in our future research.

C11: Also an addition of the 95% confidence intervals (CI 95%) of the Pearson correlations in Table 4 could support the discussion at page 16 regarding differences between correlations.

Answer: Done as you have suggested. Please see Table 4.

VERSION 2 – REVIEW

REVIEWER	Gwen McGhan University of Calgary, Canada
REVIEW RETURNED	11-Jan-2020
GENERAL COMMENTS	Thank you for the opportunity to review this revised version of the manuscript. Many of the previous comments have been

	addressed. Upon further review, here are some additional comments: On page 9 - was the response rate for the test/retest assessments the same? Did all 152 participants complete the follow up test? How was the response rate of 92.1% determined? On page 11 - a reference to support CFA within SEM may be helpful. Also on page 11 - if patients were involved in the development of the education, why are they not part of the dissemination plan? On page 14 - a reference is required for the commonly suggested cut-off level for item total correlation. Further exploration of the limitations of the study would be helpful.
--	--

REVIEWER	Roy Stewart University of Groningen/UMCG
REVIEW RETURNED	18-Dec-2019

GENERAL COMMENTS	The authors have done their best to answer my questions. I would like to thank the authors for that. But there are a few questions concerning the answers given by the authors. Major revisions: 1. The answer to the justification of the sample size is now given: "The sample size of the study was determined by the criteria suggested by Reise et al., where the intra-class correlation (ICC) >0.8.." With an intra-class correlation of >0.8, a power calculation can be performed. Moreover the sentence should be formulated in such a way that it does not create the impression that the intra-correlation >0.8 is a criterion of Reise. At the other hand the reporting: "five to ten persons for each item of the scale and sample sizes of 100 are often adequate" refers to a rule of thumb that is often applied. Applying a rule of thumb is not a power analysis. So my question is: Can the authors indicate what would be the sample size for ICC > 0.8 and what assumptions (such as type I error, type II error, etc) are used? 2. The authors state on page 11 in the improved version that all data analyses were performed using SPSS 17.0. Can the authors indicate in an appendix how the two factor model was evaluated by means of a CFA within the framework of SEM with SPSS 17.0? And can the authors indicate how the mentioned GOF indices such as NFI, CFI and RMSEA were obtained. This would be very instructive and illustrative for others who use SPSS. 3. I'm quoting from the answer: "C4: The authors say on page 8 that "No difference was found between these two methods [varimax versus oblique] in the analysis...." . How was this difference calculated? Answer: We have added some sentences to explain it in the first paragraph in the section Psychometric evaluation." This question has not been answered. The authors reported that no difference was found between the two methods. But it is not clear how this comparison was done and it is not clear what leads
---

	to the conclusion that there is no difference. Can the authors clearly indicate how the difference was calculated? Minor:  1. The Zhong study also evaluated the original P-CAT, and was also a study to validate and translate the original P-CAT version into culturally adapted version. The fact that the result of this manuscript is different from that of Zhong does not entitle the authors to claim their study, as the first study as mentioned on page 3 of article summary. 2. Page 2 add that it only concerns women in the abstract, so: Participants: 152 hospital female-staff completed the survey
--	--

VERSION 2 – AUTHOR RESPONSE

Reviewers' comments and the authors' responses:

Reviewer 1: Gwen McGhan

Thank you for the opportunity to review this revised version of the manuscript. Many of the previous comments have been addressed. Upon further review, here are some additional comments:

Comment 1 (C1): On page 9 - was the response rate for the test/retest assessments the same? Did all 152 participants complete the follow up test? How was the response rate of 92.1% determined?

AUTHORS C1: The response rate for the test/retest assessments was the same, and all 152 participants complete the follow up test. A total of 165 nurses were invited to participate in the study. Of these, 152 consented to participate, representing a response rate of 92.1%. See changes in the manuscript on page 9, lines 4-16.

C2: On page 11 - a reference to support CFA within SEM may be helpful.

AUTHORS C2: Thank you for this comment. We have added reference 36 in the manuscript after the last sentence on page 10 and to the reference list (Kline RB. Principles and Practice of Structural Equation Modeling. Fourth Edition. New York: Guilford Press 2015).

C3: Also on page 11 - if patients were involved in the development of the education, why are they not part of the dissemination plan?

AUTHORS C3: Thank you for this valuable comment. The goal of palliative care is to reduce suffering and promote quality of life for persons with progressive, incurable illnesses or injuries. The decision to use films on the patients in the education/seminars instead of older persons from the clinics have two reasons. The first reason was that older people within geriatric care in China have multi-morbidity and are in a very vulnerable condition. They are tired and have several physicals, mental, and cognitive symptoms, for example, pain and cognitive decline. That means that they have limited opportunities to contribute and share their experiences in seminars, then it takes too much energy from them. The second reason is that palliative care and death is a taboo subject in China. Therefore we mean that staff is the first target group that needs the knowledge to open up the communication and break this taboo.

C4: On page 14 - a reference is required for the commonly suggested cut-off level for item total correlation.

AUTHORS C4: The reference we used is presented in the method section: page 10, line 11. We prefer to have the results section free from references and therefore we have not included this reference on page 14.

C5: Further exploration of the limitations of the study would be helpful.

AUTHORS C5: Thank you for bringing this to our attention. We have added text in the discussion section, from page 18 line 7, about the benefit of using the original instrument and the limitations due to discriminant validity, cross-sectional design and generalization to males.

Reviewer 2: Roy Stewart

The authors have done their best to answer my questions. I would like to thank the authors for that. But there are a few questions concerning the answers given by the authors.

Major revisions:

C1. The answer to the justification of the sample size is now given: "The sample size of the study was determined by the criteria suggested by Reise et al., where the intra-class correlation (ICC) >0.8." With an intra-class correlation of >0.8, a power calculation can be performed. Moreover the sentence should be formulated in such a way that it does not create the impression that the intra-correlation >0.8 is a criterion of Reise. At the other hand the reporting: "five to ten persons for each item of the scale and sample sizes of 100 are often adequate" refers to a rule of thumb that is often applied. Applying a rule of thumb is not a power analysis. So my question is: Can the authors indicate what would be the sample size for ICC > 0.8 and what assumptions (such as type I error, type II error, etc) are used?

AUTHORS C1: The sample size of the study was determined for using a type 1 error rate at 5% and type 2 error at 80% with an expected intra-class correlation (ICC) >0.8.

C2. The authors state on page 11 in the improved version that all data analyses were performed using SPSS 17.0. Can the authors indicate in an appendix how the two factor model was evaluated by means of a CFA within the framework of SEM with SPSS 17.0? And can the authors indicate how the mentioned GOF indices such as NFI, CFI and RMSEA were obtained. This would be very instructive and illustrative for others who use SPSS.

AUTHORS C2:

Structural equation modeling (SEM) is a multivariate analysis method for exploring relations between latent constructs and measured variables. The data analyses were done with SPSS 17.0, whereas the two factor model was evaluated by means of a CFA within the framework of SEM with SPSS 22.0 under the Package 'Analysis of Moment Structures' (AMOS). It is a complex procedure to describe how to use AMOS 22.0 package to calculate NFI, CFI and RMSEA within the framework of SEM. The most important thing is that you should have AMOS 22.0 package and know how to construct the measured model (so called 'confirmatory factor analysis, CFA'). The measured model consists of observed variables and latent variables. We try to describe the whole procedure step by step:

First, install the package 'AMOS' in SPSS;

Second, double click 'Amos Graphics', you can open the AMOS window;

Third, drawing measured model by using AMOS tools. You can use AMOS tools to draw all observed variables, latent variables, paths-single headed arrows, and paths-double headed arrows, then link all observed variables by using paths-single headed arrows to a latent variable, each observed variable has a measurement error. You also can link all latent variables by using paths-double headed arrows. Observed variables are represented by rectangles, whereas latent variables are represented by ellipses.

Forth, click 'Data Files' button, select data file that you have (the file is saved as a SPSS data file), then click 'View data' button, you can look through all variables in SPSS file;

Fifth, click 'List variables in dataset' button, then link each variable (each item in questionnaire) in opened SPSS file to each observed variable in measured model, save the current path diagram as a file '*.amw';

Sixth, click 'Analysis Properties' –'Estimation' button, choose 'Maximum Likelihood', in the 'Analysis Properties' window, click 'Output' button, you can choose all parameters that you want to present in the opened window;

Finally, click 'Analyze'-'Calculate estimates' button, then click 'View the output path diagram', you can find indices of NFI, CFI and RMSEA are presented in 'Model Fit summary' window.

C3: I'm quoting from the answer: "C4: The authors say on page 8 that "No difference was found between these two methods [varimax versus oblique] in the analysis...." . How was this difference calculated?

Answer: We have added some sentences to explain it in the first paragraph in the section Psychometric evaluation."

This question has not been answered. The authors reported that no difference was found between the two methods. But it is not clear how this comparison was done and it is not clear what leads to the conclusion that there is no difference. Can the authors clearly indicate how the difference was calculated?

AUTHORS C3: We compared the numbers of items and loadings in the structural matrix by using direct varimax orthogonal rotating and oblique orthogonal rotating, and the results indicated that components in the structural matrix comprised the same numbers of items and loadings for both rotating methods.

Minor:

C4: The Zhong study also evaluated the original P-CAT, and was also a study to validate and translate the original P-CAT version into culturally adapted version. The fact that the result of this manuscript is different from that of Zhong does not entitle the authors to claim their study, as the first study as mentioned on page 3 of article summary.

AUTHORS C4: We have followed the recommendation and added a new sentence at the start on page 3. A previous study found the need for revisions of the original P-CAT questionnaire.

C5. Page 2 add that it only concerns women in the abstract, so:

Participants: 152 hospital female-staff completed the survey

AUTHORS C5: Thank you for this observation. We have added female as you suggest.

VERSION 3 – REVIEW

REVIEWER	Roy Stewart University of Groningen, The Netherlands
REVIEW RETURNED	04-Feb-2020

GENERAL COMMENTS	The authors have answered the minor revisions correctly. I would like to thank them for that. But with regard to the major revisions, I still have a number of comments/questions/revisions. Concerning C1: Sample size If the authors have used a computer program to calculate the sample size, this should be mentioned with any reference. If the authors carried out the calculations themselves using a formula(s), a reference should be given about the formula(s) used. In addition to the values already mentioned (such as type 1 error of 5%, etc.), other values used should also be mentioned. With regard to C2: SPSS, SEM/AMOS It is not clear at first reading of the manuscript that two versions of SPSS (17.0 and 22.0) have been used. If so, this should be mentioned in the manuscript. Moreover, the manuscript should also mention that the package AMOS as added to SPSS, to perform a SEM. So here it should be clearly modified in the manuscript that SPSS 17.0 and/or SPSS 22.0 has been used and the use of AMOS for the SEM part. Concerning C3: evaluation of varimax rotation relative to oblique rotation It remains unclear how the authors calculated the difference between the two methods (varimax versus oblique rotation). Maybe the authors used the eyeball test, but don't mention it. Perhaps the authors can provide the results of both varimax and oblique rotation in an appendix so that the reader can also apply the eyeball test. The authors should also correct in the manuscript that a rotation can only be orthogonal or oblique and not 'oblique orthogonal rotation'.
---

VERSION 3 – AUTHOR RESPONSE

Comments of reviewer 2, Roy Stewart

The authors have answered the minor revisions correctly. I would like to thank them for that. But with regard to the major revisions, I still have a number of comments/questions/revisions.

Concerning Comment 1: Sample size

If the authors have used a computer program to calculate the sample size, this should be mentioned with any reference.
If the authors carried out the calculations themselves using a formula(s), a reference should be given about the formula(s) used.
In addition to the values already mentioned (such as type 1 error of 5%, etc.), other values used should also be mentioned.

Our answer:

We would like to thank the reviewer for the comment. The authors have not used any formula to calculate the sample size in this study, which we used to do. Here is the setting of this psychometric study palliative care (page 8, line 21 and page 9 line 3), and the study is part of the intervention project entitled "Palliative care for elderly people with chronic illnesses: a comparative study between Sweden and China" (page 7, lines 8-10). The access to palliative care is extremely limited despite

China as one of the most populous countries, i.e., one fifth of the world's population (Yin et al. 2017, Lu et al. 2018). The limited numbers of departments in palliative care is the explanation why we included "All nurses (n=165) on duty (all shifts on the day of the data collection) at the palliative care departments of the two hospitals were deemed to be eligible for participation and were asked whether they would be willing to participate in the study" (see page 9, lines 4-6). However, we assessed before the start of the study the reasonable number to include in the sample "by the criteria suggested by Reise et al.,³³ where the intra-class correlation (ICC) >0.8 and five to ten persons for each item of the scale and sample sizes of 100 are often adequate" (see page 9, lines 8-9).

We have now clarified that the research setting for this study is palliative care (page 6, lines 8-12; and page 7, lines 8-10) and added a new reference to the reference list (Yin et al. 2017).

References:

Yin Z, Li J, Ma K, Ning X, Chen H, Fu H, Zhang H, Wang C, Bruera E, Huia D. Development of Palliative Care in China: A Tale of Three Cities. *The Oncologist* 2017;22:1362–1367.

With regard to Comment 2: SPSS, SEM/AMOS

It is not clear at first reading of the manuscript that two versions of SPSS (17.0 and 22.0) have been used. If so, this should be mentioned in the manuscript. Moreover, the manuscript should also mention that the package AMOS as added to SPSS, to perform a SEM. So here it should be clearly modified in the manuscript that SPSS 17.0 and/or SPSS 22.0 has been used and the use of AMOS for the SEM part.

Our answer:

We re-analyzed the data with SPSS 22.0, and the results in SPSS 22.0 are the same as the results in SPSS 17.0, so only SPSS 22.0 was mentioned in the manuscript. Please see page 11, line 8-10.

Concerning Comment 3: evaluation of varimax rotation relative to oblique rotation It remains unclear how the authors calculated the difference between the two methods (varimax versus oblique rotation). Maybe the authors used the eyeball test, but don't mention it. Perhaps the authors can provide the results of both varimax and oblique rotation in an appendix so that the reader can also apply the eyeball test.

Our answer:

We want to thank the reviewer for stress this issue. We used the eyeball test. The results of both varimax and oblique rotation were presented in a SPSS output file (two rotation results.spv).

Comment 4: The authors should also correct in the manuscript that a rotation can only be orthogonal or oblique and not 'oblique orthogonal rotation.'

Our answer:

We have now corrected the sentences, please see Page 9 lines 23-24 and page 10 line 2.

VERSION 4 – REVIEW

REVIEWER	Roy Stewart
-----------------	-------------

	University of Groningen, University Medical Center Groningen, Department of Health Sciences, Groningen, The Netherlands
REVIEW RETURNED	19-Mar-2020

GENERAL COMMENTS	The authors have done their best to answer the questions correctly. But with regard to page 9 lines 6- 9: “The sample size of the study was determined by the criteria suggested by Reise et al., where the intra-class correlation (ICC) > 0.8 and five to ten persons for each item of the scale and sample sizes of 100 are often adequate”, there are still ambiguities.
--

VERSION 4 – AUTHOR RESPONSE

Comments of reviewer 2, Roy Stewart:

The authors have done their best to answer the questions correctly. But with regard to page 9 lines 6-9: “The sample size of the study was determined by the criteria suggested by Reise et al., where the intra-class correlation (ICC) > 0.8 and five to ten persons for each item of the scale and sample sizes of 100 are often adequate”, there are still ambiguities.

Our answer:

We agree that the text after the previous revision was still unclear. The new sentence in the method section is more straightforward (see page 9, lines 6-7), and we have added text in the discussion (see page 19, lines 18-23), as well as four new references in the reference list. Palliative care is not routine health care at the hospitals in China, and this means that the number of nurses in palliative care is limited to a few hospitals in Yunnan province. It is essential to making the psychometric test on the group of staff that are included in the ongoing project, i.e. nurses in palliative care.